# Effect of Novel AKT Inhibitor Vevorisertib as Single Agent and in Combination with Sorafenib on Hepatocellular Carcinoma in a Cirrhotic Rat Model

**DOI:** 10.3390/ijms232416206

**Published:** 2022-12-19

**Authors:** Keerthi Kurma, Ayca Zeybek Kuyucu, Gaël S. Roth, Nathalie Sturm, Marion Mercey-Ressejac, Giovanni Abbadessa, Yi Yu, Herve Lerat, Patrice N. Marche, Thomas Decaens, Zuzana Macek Jilkova

**Affiliations:** 1Institute for Advanced Biosciences, University Grenoble Alpes, CNRS UMR5309, INSERM U1209, 38700 Grenoble, France; 2Hepato-Gastroenterology and Digestive Oncology Department, CHU Grenoble Alpes, 38700 Grenoble, France; 3Pathology and Cytology Department, CHU Grenoble Alpes, 38700 Grenoble, France; 4T-RAIG, TIMC, University Grenoble-Alpes/CNRS UMR5525, 38700 La Tronche, France; 5ArQule Inc., Burlington, MA 01803, USA; 6Unité Mixte de Service hTAG, Grenoble Alpes University, Inserm US046, CNRS UAR2019, 38700 La Tronche, France

**Keywords:** HCC, DEN-induced cirrhotic rat model of HCC, liver, fibrosis, vevorisertib, AKT pathway

## Abstract

Hepatocellular carcinoma (HCC) is one of the leading causes of cancer-related mortality worldwide. The AKT pathway is often activated in HCC cases, and a longer exposure to tyrosine kinase inhibitors such as sorafenib may lead to over-activation of the AKT pathway, leading to HCC resistance. Here, we studied the efficacy of a new generation of allosteric AKT inhibitor, vevorisertib, alone or in combination with sorafenib. To identify specific adverse effects related to the background of cirrhosis, we used a diethylnitrosamine (DEN)-induced cirrhotic rat model. Vevorisertib was tested in vitro on Hep3B, HepG2, HuH7 and PLC/PRF cell lines. Rats were treated weekly with intra-peritoneal injections of DEN for 14 weeks to obtain cirrhosis with fully developed HCC. After that, rats were randomized into four groups (*n* = 7/group): control, sorafenib, vevorisertib and the combination of vevorisertib + sorafenib, and treated for 6 weeks. Tumor progression was followed by MRI. We demonstrated that the vevorisertib is a highly potent treatment, blocking the phosphorylation of AKT. The tumor progression in the rat liver was significantly reduced by treatment with vevorisertib + sorafenib (49.4%) compared to the control group (158.8%, *p* < 0.0001). Tumor size, tumor number and tumor cell proliferation were significantly reduced in both the vevorisertib group and vevorisertib + sorafenib groups compared to the control group. Sirius red staining showed an improvement in liver fibrosis by vevorisertib and the combination treatment. Moreover, vevorisertib + sorafenib treatment was associated with a normalization in the liver vasculature. Altogether, vevorisertib as a single agent and its combination with sorafenib exerted a strong suppression of tumor progression and improved liver fibrosis. Thus, results provide a rationale for testing vevorisertib in clinical settings and confirm the importance of targeting AKT in HCC.

## 1. Introduction

Hepatocellular carcinoma (HCC) is currently one of the most common cancers and also one of the leading causes of cancer deaths worldwide, accounting for more than 830,000 deaths each year [1]. The similarity between HCC incidence and mortality clearly shows that the prognosis associated with this type of cancer is very poor. Thus, HCC represents today a major global healthcare challenge. The main risk factors for HCC development are chronic hepatitis infection by the hepatitis B virus (HBV) or hepatitis C virus (HCV), aflatoxin, alcoholic liver disease and metabolic liver diseases. The distribution of these risk factors depends on geographic regions and race/ethnic groups, and they are highly variable among patients with HCC. However, the connecting element is the fact that most patients develop HCC on the background of chronic liver disease characterized by sustained inflammatory damage, abnormal vasculature, hepatocyte necrosis and regeneration and associated fibrotic deposition [1]. Indeed, 80–90% of HCC is associated with the formation and progression of fibrosis and cirrhosis.

For more than a decade, systemic molecular therapies, such as sorafenib, have been the backbone treatment of advanced-stage HCC. However, they showed only a modest improvement in the overall survival of HCC patients, with side effects altering the quality of the patient’s life [1]. Today, immunotherapies such as immune-checkpoint inhibitors are revolutionizing the management of HCC, and the combination of an anti-PD-L1 (atezolizumab) plus an anti-VEGFA (bevacizumab) and an anti-PD-L1 (durvalumab) plus an anti-CTLA4 (tremelilumab) have shown superior efficacy compared to sorafenib in advanced-stage patients. However, recent data show that around 70–80% of HCC patients treated by this type of combination do not respond to the therapy, and their survival prognosis remains poor [2]. Therefore, several emerging combinations of various systemic therapies are under clinical trials and other possibilities are currently being studied.

AKT has been proposed as a therapeutic target in oncology for decades. In fact, the PI3K/AKT/mTOR pathway is a major intracellular signaling pathway that is involved in the regulation of cell proliferation, growth and survival. In the mechanism of activation of the PI3K/AKT/mTOR pathway, the membrane lipid phosphatidylinositol 4, 5-bisphosphate (PIP2) is phosphorylated by phosphatidylinositol 3-kinase into phosphatidylinositol 3, 4, 5-triphosphate (PIP3), which then binds to and activates the serine/threonine kinase AKT [3,4]. Once activated, the AKT activates downstream signaling effectors to regulate cell survival, proliferation, cell cycle progression, migration and angiogenesis [5,6]. Nearly 50% of patients with HCC have shown the dysregulation of the AKT/mTOR pathway, which may be partially associated with activation signals from receptor tyrosine kinases, such as IGFR and/or EGFR pathways [7]. A study conducted by Zhou et al. demonstrated that the PI3K/AKT/mTOR pathway is more significantly activated in high-grade HCC tumors and is associated with poor prognosis in HCC patients [8], suggesting that the activation of the PI3K/AKT/mTOR pathway functionally contributes to HCC progression. Recent studies revealed that long-term treatment with sorafenib leads to AKT upregulation, and the inhibition of the PI3K/AKT signaling pathway can reverse sorafenib-derived resistance in HCC [9,10,11,12].

Currently, allosteric and catalytic AKT inhibitors are being investigated for HCC treatments, as was updated recently [13]. Among them, the allosteric inhibitor ARQ 092/MK-7075/miransertib is particularly interesting, since it binds to both the active and inactive forms of AKT [14]. We previously reported that miransertib exhibited anti-proliferative activity and strong anti-tumor activity in an in vivo model of carcinogen-induced HCC [15,16]. Importantly, the next-generation AKT inhibitor ARQ 751/MK-4440/vevorisertib, tested in this study, displays enhanced pharmacokinetic properties and potency compared to miransertib [5].

We postulate that therapy with vevorisertib will be able to treat a fully developed HCC by inhibiting the PI3K/AKT/mTOR pathway and may prevent/overcome the sorafenib resistance in HCCs. Therefore, the combination of vevorisertib and sorafenib can represent a therapeutic strategy that will improve treatment effectiveness in HCC. To identify specific adverse effects that could be related to the background of cirrhosis, this therapeutic strategy should be pre-clinically tested in an appropriate animal model. As fibrosis/cirrhosis modifies liver vascularization, extracellular matrix composition and drug metabolism, it is essential to use a cirrhotic animal model to assay HCC drugs in order to check their efficacy on tumors but also tolerance to the treatment. Hence, we used the DEN-induced cirrhotic rat model of HCC that reproduces key features of human HCC [17] to attest the safety and prove the efficacy of AKT inhibitor vevorisertib as a single treatment and in combination with sorafenib.

## 2. Results

### 2.1. In Vitro Experiments

First, we determined the concentrations of vevorisertib that induced 20% and 50% inhibition (IC20 and IC50) on HCC cell lines Hep3B, HuH7 and PLC/PRF/5 and on hepatoblastoma cell line HepG2, as seen in Appendix A. We chose these cell lines because of the different genetic profiles including TP53 mutation (HuH7 and PLC/PRF/5), TP53 deletion (Hep3B), beta-catenin mutation (HepG2), normal *p*-AKT (Hep3B) vs. low *p*-AKT (HepG2, HuH7, PLC/PRF/5). The potency ratio between sorafenib IC50 and vevorisertib IC50 showed the high effectiveness of vevorisertib compared to sorafenib. The Western blot analyses indicated that vevorisertib treatment completely blocked AKT (Ser473) phosphorylation in all tested cell lines, both at IC20 and IC50 concentrations, as seen in Appendix A, as well as in a combination setting, as seen in Appendix A.

### 2.2. In Vivo Experiments

To investigate the efficacy of vevorisertib, rats were treated for six weeks by vehicle (control group), sorafenib, vevorisertib or vevorisertib + sorafenib using a DEN-induced cirrhotic rats model of HCC obtained after 14 weeks of DEN-injections, Figure 1. The AKT inhibitor was administered with respect to the optimal schedule “5 days on and 9 days off”, as published previously [15].

#### 2.2.1. Safety Assessment

No significant effect on body weight or liver weight was observed at the end of the treatment, as seen in Table 1. Similarly, we did not find any difference in food consumption.

Assessment of the intrahepatic triglycerides did not show any difference between groups. Blood sample analyses revealed that none of the treatments affect triglyceride concentration, glucose levels or prothrombin time compared to the control treatment. Similarly, the levels of alkaline phosphatase (ALP) did not differ between the groups. On the other hand, serum levels of albumin, aspartate aminotransferase (AST), alanine aminotransferase (ALT) and gamma-glutamyl transpeptidase (GGT) were significantly lower in the vevorisertib + sorafenib group compared to control and sorafenib. Moreover, the serum levels of albumin, ALT and GGT were significantly lower in the vevorisertib + sorafenib group compared to sorafenib treatment alone. We also observed significantly lower total bilirubin in vevorisertib alone and the vevorisertib + sorafenib group compared to the control. Cholesterol was higher in vevorisertib alone compared to vevorisertib + sorafenib or control. No difference in the clinical and biological analysis was observed between the sorafenib treated group and the control group.

Altogether, our results showed that vevorisertib + sorafenib treatment improves liver function without affecting glucose or triglyceride levels.

#### 2.2.2. Effect on Tumor Progression

The tumor progression was followed by a liver MRI scan during the 6 weeks of the treatment. We observed that tumor progression was reduced in all treated groups compared to the control group but reached a significant difference only in the vevorisertib + sorafenib group (Control: 158.8 ± 11.6% vs. vevorisertib + sorafenib group: 49.4 ± 5.12%, *p* < 0.0001), Figure 2a,b.

By macroscopic examination of the liver, we observed that the mean tumor size was significantly reduced in vevorisertib (4.3 ± 0.4 mm, *p* = 0.0055) and in vevorisertib + sorafenib (3.3 ± 0.2 mm, *p* < 0.0001) compared to the control group (9.9 ± 0.9 mm), Figure 2c,d. In addition, the mean tumor size in the vevorisertib + sorafenib group was significantly reduced when compared with sorafenib treatment alone (*p* = 0.0116). Similarly, the number of tumors on the liver surface was significantly lower in rats treated by vevorisertib and vevorisertib + sorafenib compared to control. In fact, the tumor number was decreased by 63% in rats treated by vevorisertib (36.6 ± 8.2; *p* = 0.0163) and by 81% in rats treated by vevorisertib + sorafenib (18.2 ± 2.8; *p* < 0.0001) compared to the control group (100.4 ± 7.3), Figure 2e. The vevorisertib + sorafenib group displayed a significantly lower number of tumors compared to sorafenib-treated animals (*p* = 0.0228).

Hence, vevorisertib and vevorisertib + sorafenib treatments showed significant efficacy in the control of tumoral progression.

#### 2.2.3. Effect on Tumor Cell Proliferation

Accordingly, the frequencies of Cyclin D1-positive nuclei in tumor area were significantly lower in vevorisertib (20.1 ± 4.0, *p* = 0.0420) and in the vevorisertib + sorafenib group (16.1 ± 3.1, *p* = 0.0073) compared to the control group (58.9 ± 7.6), Figure 3a,b. Similarly, the frequency of Ki67+ tumor cells was significantly decreased in vevorisertib (9.3 ± 1.1, *p* = 0.0293) and the vevorisertib + sorafenib group (6.0 ± 0.8, *p* = 0.0007) groups compared to the control group (43.7 ± 11.3), Figure 3c,d. No significant reduction in tumor cell proliferation was observed between sorafenib and the control group. Thus, results of Ki67 and Cyclin D1 staining demonstrated that vevorisertib and vevorisertib + sorafenib treatments reduce HCC proliferation.

#### 2.2.4. Effect of Treatment on Tumor Vascularization and Liver Fibrosis

Next, we used rat-specific anti-CD34 antibody to evaluate the effect of treatment on vascularization. The liver tissue of the DEN-induced model of HCC is characterized by numerous structural abnormalities in the vasculature [17], as observed in the control animals, Figure 4a.

Significant improvement and a normalization of vasculature was observed in the vevorisertib + sorafenib treated groups, as confirmed by the quantification of vascular density, Figure 4b.

Liver fibrosis was analyzed by Sirius red staining, Figure 4c. The collagen network was significantly reduced in vevorisertib (55.1 ± 6.6%, *p* = 0.0352) and the vevorisertib + sorafenib (45.1 ± 3.4%, *p* = 0.0013) group compared to the control group (100 ± 10.7%), Figure 4d. Additionally, the vevorisertib + sorafenib combination treatment also significantly reduced fibrosis compared to the single sorafenib-treated group (*p* = 0.0131). Furthermore, the effect of vevorisertib and vevorisertib + sorafenib treatment on liver fibrosis was confirmed by qPCR analysis, Figure 4e. The expression of fibrosis marker α-smooth muscle actin (*α-SMA*) was significantly downregulated in liver tissue in the vevorisertib and vevorisertib + sorafenib groups compared to the control group (*p* = 0.0043, *p* < 0.0001). Similarly, when compared to sorafenib single treatment, the vevorisertib + sorafenib group exerted significantly reduced gene expression of *α-SMA* (*p* = 0.0426), Figure 4e. The expression of collagen 1 (*COL1*) was significantly reduced in the vevorisertib + sorafenib group compared to the control group (*p* = 0.0010). As expected, the expression of the tissue inhibitor of metalloproteinases 1 (*TIMP1*) was decreased by vevorisertib + sorafenib treatment, while matrix metalloproteinase-2 (*MMP-2*) and matrix metalloproteinase-9 (*MMP-9*) were increased compared to the control group. Finally, the vevorisertib + sorafenib treatment reduced the expression of transforming growth factor (*TGF-β*) in liver tissue.

Altogether, vevorisertib and, mainly, the vevorisertib + sorafenib combination treatment significantly improve liver fibrosis in a DEN-induced cirrhotic model of HCC.

## 3. Discussion

Despite clear signs of anticancer activity in vitro and in pre-clinical studies, AKT-targeting agents have reached limited success in the clinic. The recent systematic investigation of available AKT inhibitors revealed that allosteric AKT inhibitors may partially block some of the downstream effector kinases of AKT. On the contrary, ATP-competitive inhibitors may cause their activation possibly through feedback relief, which limits their efficacy [18].

Importantly, recent studies revealed that AKT inhibitor MK2206 significantly increases the sensitivity of HCC cells to sorafenib in vitro [12], and similarly, the inhibition of AKT signaling can sensitize HCC to Lenvatinib treatment [19]. Thus, targeting AKT pathways together with sorafenib and other treatments of HCC is a promising therapeutic strategy.

In this study, we carried out an evaluation of the effect of novel allosteric AKT inhibitor vevorisertib as a single agent and in combination with sorafenib on HCC in a cirrhotic rat model to define its anti-tumor activity and the safety in an animal model that reproduces the key features of human HCC, including fibrosis/cirrhosis [17]. It is important to note that the combination of vevorisertib + sorafenib has never been used in clinical practice and has never even been tested in pre-clinical models of CHC.

Our work demonstrates that vevorisertib is highly potent drug reducing HCC proliferation and tumor progression. Additionally, vevorisertib was efficacious in the modulation of the liver microenvironment. This is particularly important as the tumor microenvironment plays a crucial role in tumor progression. Indeed, AKT inhibitors are known to directly affect cancer cells but also exhibit indirect anti-tumor activity mediated by the modulation of the tumor microenvironment [20].

Increased and irregular vasculature allows small HCC lesions to progress and metastasize, which is a typical situation in the fibrotic liver characterized by a constant increase in blood vessel formation [21]. Here, we showed that vevorisertib in combination with sorafenib normalize the vascularization of liver tissue. The anti-fibrotic effect of sorafenib was previously demonstrated [22]; however, in our study, sorafenib treatment did not significantly improve the fibrotic status of the liver. We observed an important amelioration of liver fibrosis after vevorisertib and the vevorisertib + sorafenib combination treatment compared to control, which was confirmed by a shift in the matrix regulatory pathway leading to fibrosis resolution with a strong decrease in collagen accumulation. Another essential determinant of HCC progression and survival is cancer-associated inflammation, with TGFβ orchestrating a favorable microenvironment for tumor progression [23]. Here, we showed that expression TGFβ in liver tissue was downregulated by the vevorisertib + sorafenib combination. Moreover, it was previously demonstrated that TGFβ activates the process of fibrogenesis in an AKT dependent manner and that the fibrosis in a mouse model can be reduced by the PI3K/mTOR inhibitor [24]. Thus, we assume that the additive effect of the sorafenib and vevorisertib on the reduction of TGFβ expression contributes to the reduction of fibrosis and to the normalization of vasculature in the vevorisertib + sorafenib treated group.

In summary, we provide evidence that the allosteric AKT inhibitor vevorisertib, alone or in combination with sorafenib, potently inhibits the AKT pathway. Despite difficult conditions with an aggressive model of cancer in cirrhotic rats, a single vevorisertib treatment and the combination of vevorisertib + sorafenib exhibit efficacy in controlling tumor progression and demonstrate a good safety profile that makes this experimental drug relevant in the treatment of HCC in cirrhotic patients. In conclusion, the results presented here confirm the importance of targeting AKT in HCC development and progression. The high potency of vevorisertib warrants further clinical investigation in patients with HCC.

## 4. Materials and Methods

### 4.1. In Vitro Experiments

We used three different human HCC cell lines: Hep3B, HuH7 and PLC/PRF/5, and one hepatoblastoma cell line: HepG2. These cell lines are not characterized by AKT mutations, but the *p*-AKT expression differs between cell lines with normal expression in Hep3B and is low in HepG2, HuH-7 and PLC/PRF/5 cell lines [25]. HepG2 and Hep3B cells were cultured in Minimum Essential Medium (MEM, GIBCO™, Life Technologies, Renfrew, UK), with GlutaMAX™ supplement. Huh-7 cells were incubated in Dulbecco’s Modified Eagle Medium (DMEM, GIBCO™, Life technologies) with GlutaMAX™ supplement. PLC/PRF/5 cells were cultured in DMEM supplemented with 1% sodium pyruvate. Mediums were supplemented with 10% fetal bovine serum and 1% antibiotics (Pen Strep, Life Technologies). An MTT (3-(4,5-Dimethylthiazol-2-yl)-2,5-diphenyltetrazolium bromide) assay was used for cell viability testing.

### 4.2. Treatment

Vevorisertib was obtained from ArQule Inc., Burlington, MA, USA, and sorafenib for the in vitro study was purchased from Sigma-Aldrich, Germany (Bay 43-9006) and, for the in vivo study, from Bayer HealthCare, Germany (Nexavar). For the in vitro experiment, a fresh solution of the treatment was prepared every week and stored at room temperature, protected from light. The sorafenib treatment was prepared as described previously [15,16]. Briefly, the sugar coating of 200 mg sorafenib tablets was dissolved in DMSO. Sorafenib was mixed with 1 mL of poly-oxyl castor oil (Cremophor^®^ EL, Sigma-Aldrich) and 1 mL of 95% ethanol per tablet to emulsify and solubilize it. The emulsion was then diluted in purified water to obtain a 10 mg/mL solution of Sorafenib suitable for oral gavage. The dose strategy for vevorisertib was based on a previous toxicity study. Vevorisertib was dissolved in a 0.01 M phosphoric acid solution to obtain a 10 mg/mL solution suitable for oral gavages with a final pH of 2.25 ± 0.15. The combination was prepared by mixing drugs just before oral gavages.

### 4.3. Animals and Treatment Protocol

Six-week-old Fischer 344 male rats (Charles River Laboratories, France) were housed in the animal facility of Plateforme de Haute Technologie Animale (Jean Roget, University of Grenoble Alpes, Grenoble, France). The rats were kept in individually ventilated cage (IVC) systems at constant temperature and humidity with 3 animals per cage. Intra-peritoneal injections of 50 mg/kg DEN diluted in pure olive oil (Sigma-Aldrich, Steinheim am Albuch, Germany) were administered once per week in order to obtain a fully developed HCC on a cirrhotic liver after 14 weeks [17,26]. After 14 weeks of DEN injections, rats were randomized in four different groups (*n* = 7 rats/group) as follows: control (vehicle-treated) group, sorafenib group, vevorisertib group and vevorisertib + sorafenib. Treatments were administered by daily oral gavage for a period of six weeks. The vevorisertib treatment was administered 5 days on—9 days off—5 days on—9 days off—5 days on—9 days off, with a total of 15 days of treatment, at a dose 10 mg/kg/day, as recommended by ArQule Inc. Animals were killed after the final “9 days off” period. Sorafenib was administered continuously at a dose of 10 mg/kg/day every day. To perform oral gavages and MRI analyses, the rats were transported to the Grenoble Institute of Neuroscience (GIN, INSERM, U1216, University of Grenoble-Alpes, France) equipped by Grenoble MRI facility IRMaGE. During treatment, all rats were daily weighed to monitor the nutritional state and to adapt treatment doses. Protein-rich nutrition was added to the standard food in cages when a loss of weight was observed. All animals received humane care in accordance with Guidelines on the Humane Treatment of Laboratory Animals.

### 4.4. MRI Analyses

All rats from this project were subjected to three MRI scans. MRI1 was performed before randomization. MRI2 and MRI3 were performed after three weeks and after six weeks of treatment. MRI scans imaging study was performed with a 4.7 Tesla MR Imaging system (BioSpec 47/40 USR, Bruker Corporation, Germany) and Transmit/Receive Volume Array. Coil for rat body 8 × 2 (Bruker Corporation, Germany) in the Grenoble MRI facility IRMaGE. Rats were fitted in the ventral decubitus position and anesthetized with isoflurane inhalation (Forane^®^, Abbott, Chicago, IL, USA); breathing was continuously monitored to maintain a respiratory rate between 35 and 45 breaths per minute, and body temperature was maintained around 37 °C. We used the Turbo rapid acquisition with relaxation enhancement T2-weighted (Turbo-RARE T2) sequence (repetition time (TR): 1532.9 ms, echo time (TE): 27.4 msec, flip angle (FA): 180°). MRI parameter adjustment and image acquisition were realized by using Paravision 5.1 software. A morphological analysis was realized based on the TurboRARE T2 sequences and according to the response evaluation criteria in solid tumors (RECIST) criteria. The five largest liver tumors per each rat were selected and measured on MRI; estimated tumor size corresponded to the sum of the diameter of these 5 largest lesions. For each rat, MRI1 was considered as the baseline (i.e.: 0%), and tumor progression was calculated as tumor size (MRI3–MRI1)/MRI1.

### 4.5. Morphological and Histopathological Analyses

Rats were euthanized under randomly fed conditions, with abdominal aorta blood sampling for hematologic and biochemical analyses. The serum was tested for liver safety markers: glucose, alkaline phosphatase (ALP), alanine transaminase (ALT), aspartate transaminase (AST), total bilirubin, albumin, cholesterol, and gamma glutamyltransferase (GGT) by Charles River Clinical Pathology Services using Olympus instruments. The rats’ organs were weighed. The liver organ from each rat was weighed, the visible nodules (larger than 1 mm) on the liver surface were counted, and the diameter of the five biggest nodules was measured. The mean of these 5 diameters was calculated to obtain a histopathological estimation of the tumor size per each rat.

### 4.6. Immunohistochemical and Immunofluorescence Analyses

For histological analysis, liver samples were fixed in neutral buffered 10% formalin solution (Sigma-Aldrich, Steinheim am Albuch, Germany) and paraffin-embedded. Four-micrometer sections of liver tissue were prepared. To detect proliferating cells, paraffin-embedded sections were incubated overnight at 4 °C with the primary antibody anti-Cyclin D1 (Abcam, EPR2241, dilution 1:200) or with anti-Ki67 (Thermofisher scientific, SP6, dilution 1:150), followed by incubation with the anti-rabbit EnVision system HRP Labelled Polymer (Dako Agilent, USA). DAB was used as the chromogen for immune detection. Positively stained cells were quantified using ImageJ software (NIH, USA) on 10–15 randomly selected fields/sections (20× magnification), captured by an Olympus BX41 microscope. Data are presented as percentage of cells with positive nuclei per total cells (HPF: high-power fields; 20× magnification). To detect vascularization, the sections were incubated overnight at 4 °C with an anti-rat CD34 antibody (R&D systems, polyclonal, dilution 1:100), followed by incubation with Alexa 647-conjugated donkey anti-goat IgG (Life Technologies, Carlsbad, CA, USA). Images were captured using the ApoTome microscope (Carl Zeiss, Germany) equipped with a camera AxioCam MRm and collected by AxioVision software. The positive area threshold was quantified using ImageJ software (NIH, MD, USA) on 10 randomly selected fields/sections (10× magnification). Collagen was detected on paraffin-embedded sections with picro-sirius red stain solution (Sigma-Aldrich). The positively stained area was quantified using ImageJ software (NIH, MD, USA) on 15 randomly selected fields/sections (10× magnification) captured by an Olympus BX41 microscope. Data were presented as a positive area. All analyses were performed in a double-blinded manner.

### 4.7. Real Time Polymerase Chain Reaction (qPCR)

Total RNA was extracted from rat liver tissue samples preserved with an RNA stabilization solution (Thermo scientific, MA, USA). RNA purification was performed with RNeasy Mini Kit^®^ (Qiagen, USA). Reverse transcription was realized with iScriptTM Reverse transcription supermix Kit (BioRad, CA, USA), and amplification reactions were performed in a total volume of 20 µL by using a Thermocycler sequence detector (BioRad CFX96, CA, USA) with qPCR kit iTaqTM Universal SYBR^®^Green Super mix (BioRad, CA, USA). GADPH was used as a housekeeping gene. The primers, listed in Table 2, were designed with Primer 3 software (version 4.0.0) and verified on BLAST. The oligonucleotide sequences were synthesized by Eurofins Genomics ^®^ in 0.01 µmol scale, with a salt-free level of purification. Every analysis was performed in duplicate.

### 4.8. Immunoblot Analysis

Liver homogenates were prepared in EZ buffer (20 mM Tris; 100 mM NaCl; EDTA 1 mM; 0.5% NP40; 10% glycerol; 1X anti phosphatase and 1X protease) containing proteins, which were then denatured in Laemmli sample buffer (Bio-Rad) containing 5% β-mercaptoethanol and separated by gel electrophoresis (Mini Protean Gels ^®^, Bio-Rad, Hercules, CA, USA) and transferred to nitrocellulose (Bio-Rad) membranes using a wet blot method. The membranes were blocked in TBS-Tween solution with 5% BSA for 1 h at 4 °C.

### 4.9. Statistical Analysis

The comparisons of means were calculated by using ANOVA tests with Tukey HSD correction for multiple means comparisons and independent T-tests only when two means were compared. Data are presented as mean values ± standard error mean (SEM). The statistical analyses were performed using Prism 9 (GraphPad Software Inc., San Diego, CA, USA).

## Figures and Tables

**Figure 1 ijms-23-16206-f001:**
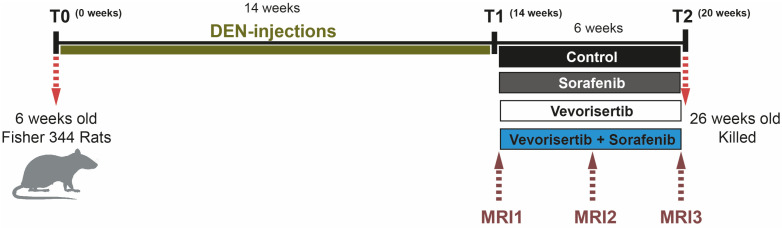
The schema of the study evaluating the effect of AKT inhibitor vevorisertib as a single agent and in combination with sorafenib in a DEN-induced cirrhotic rat model of HCC.

**Figure 2 ijms-23-16206-f002:**
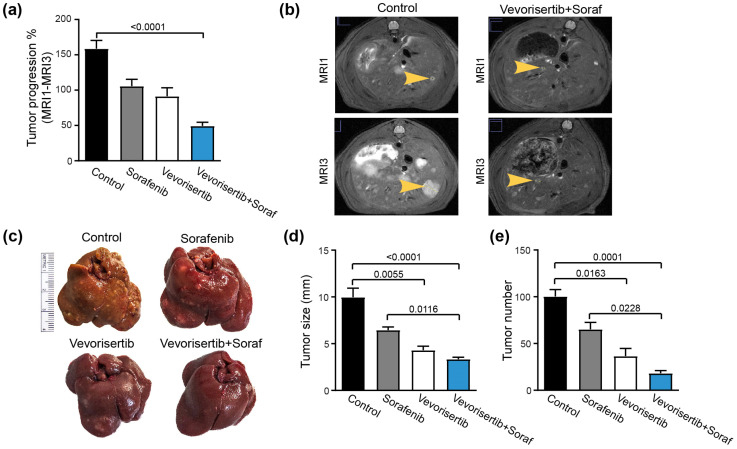
Effect of vevorisertib and vevorisertib + sorafenib treatment on tumor progression. (**a**) Tumor progression assessment calculated as tumor size (MRI3–MRI1)/MRI1 and (**b**) representative images of abdominal MRI1 and MRI3 scans of control and vevorisertib + sorafenib treated rats. (**c**) Representative images of liver of control, sorafenib, vevorisertib and vevorisertib + sorafenib treated group. (**d**) Macroscopic examination of livers with an assessment of tumor size calculated as average diameter of the five largest tumors. (**e**) Macroscopic examination of tumor number at the surface of the livers. *n* = 7/group, values are mean ± SE. Kruskal–Wallis test was used for multiple comparisons.

**Figure 3 ijms-23-16206-f003:**
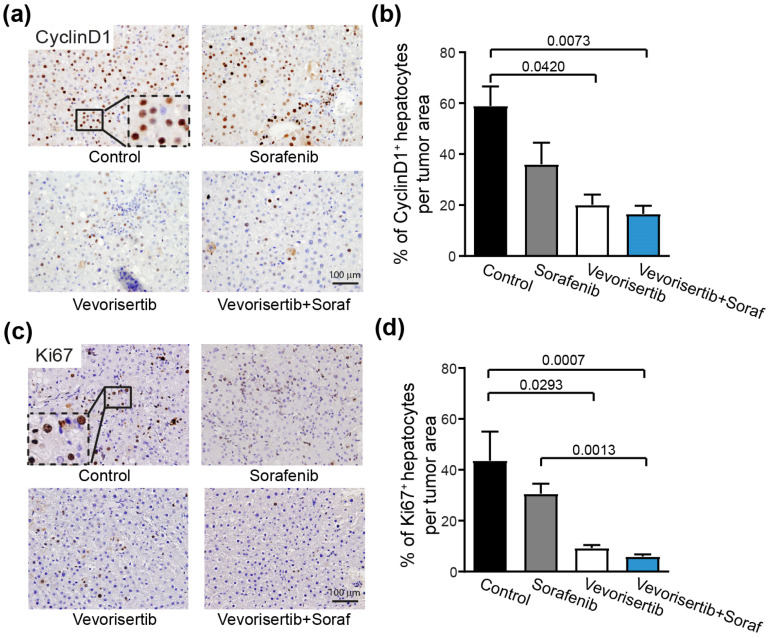
Effect of vevorisertib and vevorisertib + sorafenib treatment on tumor cell proliferation. (**a**) Representative images of nuclear Cyclin D1 staining of control, sorafenib, vevorisertib and vevorisertib + sorafenib treated group, 20× magnification and (**b**) quantification of % of Cyclin D1 positive hepatocytes per tumor area. (**c**) Representative images of nuclear Ki67 staining of control, sorafenib, vevorisertib and vevorisertib + sorafenib treated group, 20× magnification and (**d**) quantification of % of Ki67 positive hepatocytes per tumor area. *n* = 7/group, values are mean ± SE. Kruskal–Wallis test was used for multiple comparisons.

**Figure 4 ijms-23-16206-f004:**
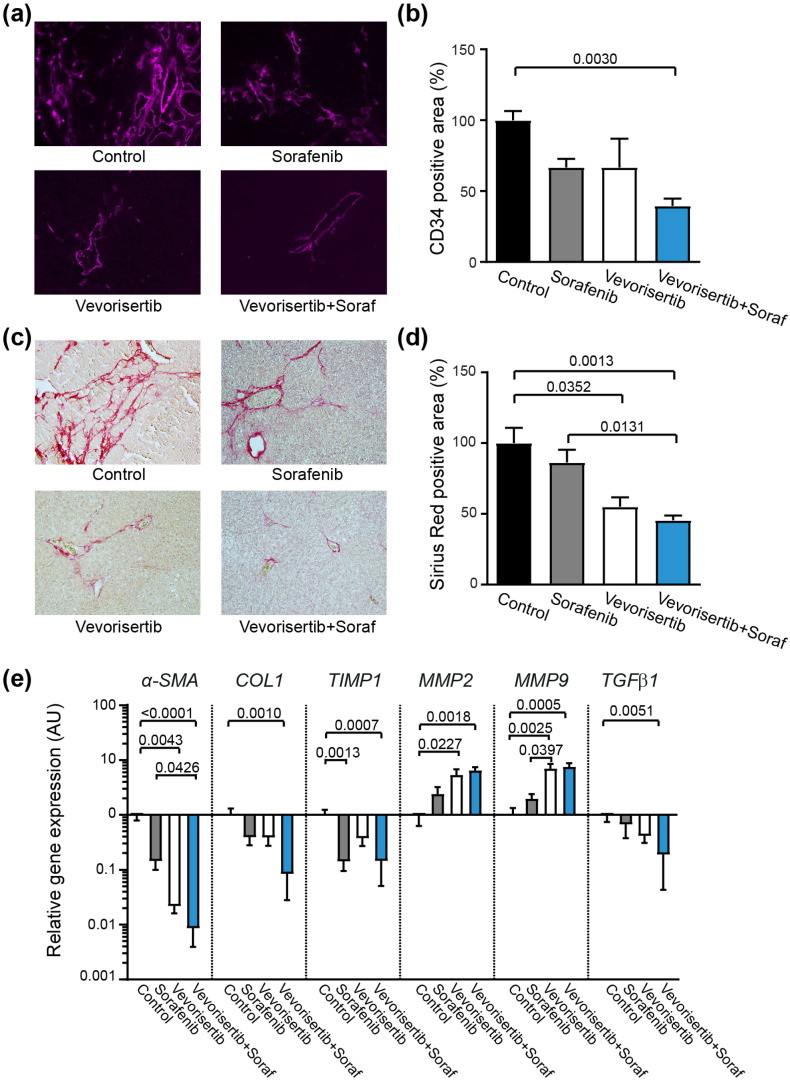
Effect of vevorisertib and vevorisertib + sorafenib treatment on tumor vascularization and liver fibrosis. (**a**) Representative images of CD34 immunofluorescence staining of control, sorafenib, vevorisertib and vevorisertib + sorafenib treated group, 20× magnification, (**b**) quantification of CD34 positive staining area, mean of control group was set as 100%. (**c**) Representative images of liver tissues stained with Sirius red, 20× magnification, (**d**) quantification of Sirius red staining per area, mean of control group was set as 100%. (**e**) Relative gene expression of alpha-smooth muscle tissue (*α-SMA*), collagen 1 (*COL1*), tissue inhibitor of metalloproteinases 1 (*TIMP1*), matrix metalloproteinase-2 (*MMP-2*), matrix metalloproteinase-9 (*MMP-9*) and transforming growth factor (*TGF-β*) in liver tissue; mean of control group was set as 1. *n* = 7/group; values are mean ± SE. The Kruskal–Wallis test was used for multiple comparisons.

**Table 1 ijms-23-16206-t001:** Clinical and biological analysis.

	Control	Sorafenib	Vevorisertib	Vevorisertib + Soraf	*p*-Value
Body Weight (g)	291 ± 5.0	290 ± 4.1	290 ± 3.3	281 ± 3.3	0.1520
Liver					
Weight (g)	13.0 ± 0.4	13.1 ± 0.3	13.1 ± 0.4	11.6 ± 0.6	0.0567
TG (g/L)	31.5 ± 4.9	26.6 ± 1.9	28.5 ± 2.5	26.3 ± 1.7	0.5783
Blood					
Albumin (g/dL)	3.5 ± 0.18	3.6 ± 0.06	3.6 ± 0.03	3.3 ±0.05 *^, #^	0.0127
AST (U/L)	97.4 ± 5.9	90.2 ± 4.8	82.9 ± 2.9	75.3 ± 3.6 *	0.0283
ALT (U/L)	69.3 ± 3.9	71.2 ± 3.4	61.8 ± 3.4	50.3 ± 3.0 *^, ##^	0.0029
ALP (U/L)	199.2 ± 9.7	212.0 ± 7.6	184.6 ± 7.1	192.0 ± 5.7	0.1499
GGT (U/L)	14.0 ± 3.0	15.3 ± 2.1	7.7 ± 1.7	2.7 ± 1.1 *^, ##^	0.0013
PT (s)	18.1 ± 0.5	19.2 ± 1.3	18.0 ± 0.3	18.5 ± 0.5	0.9013
Total Bilirubin (mg/dL)	0.25 ± 0.04	0.18 ± 0.01	0.15 ± 0.01 *	0.14 ± 0.01 ***	0.0002
Creatinine (mg/dL)	0.37 ± 0.02	0.33 ± 0.02	0.40 ± 0.01	0.31 ± 0.01	0.0700
GLU (mg/dL)	132.3 ± 7.8	141.1 ± 5.7	152.6 ± 7.0	140.0 ± 6.2	0.2618
Cholesterol (mg/dL)	84.4 ± 4.3	86.2 ± 3.0	102.6 ± 2.0 *^, &^	84.7 ± 2.2	0.0044
TG (g/L)	78.4 ± 11.1	70.3 ± 10.7	54.0 ± 11.5	60.6 ± 13.1	0.4764

AST, aspartate aminotransferase; ALT, alanine aminotransferase; ALP, alkaline phosphatase; GGT, gamma-glutamyl transpeptidase; PT, prothrombin time; GLU, glucose; TG, triglycerides. Values are means ± SE. Significant difference compared to control; *: *p* < 0.05; ***: *p* < 0.001. Significant difference compared to sorafenib; #: *p* < 0.05; ##: *p* < 0.01. Significant difference between vevorisertib + sorafenib and vevorisertib alone; &: *p* < 0.05. *n* = 7/group. The comparable biological analysis of 6 weeks old rats at T0 before DEN injections are: Albumin 3.23 ± 0.04 g/dL; AST 82.2 ± 8.68 U/L; ALT 53.2 ± 1.9 U/L; Total Bilirubin 0.17 ± 0.02 mg/dL; Creatinine 0.20 ± 0.00 mg/dL ([17]).

**Table 2 ijms-23-16206-t002:** List of primer sequences for qPCR analysis.

Gene	Reverse Sequence (5′–3′)	Forward Sequence (5′–3′)
α-SMA	CATCTCCAGAGTCCAGCACA	ACTGGGACGACATGGAAAAG
COL1	CTTCTGGGCAGAAAGGACAG	GCCAAGAAGACATCCCTGAA
TGF-β1	TGGGACTGATCCCATTGATT	ATACGCCTGAGTAGCTGTCT
TIMP1	TGGCTGAACAGGGAAACACT	CAGCAAAAGGCCTTCGTAAA
MMP2	GGGTTTCTTCTGGCTCAGG	TCTGGCTATCCACAAGACTGG
MMP9	GGAAAAGGAAGGAGGGTACG	CCACTCAGGGCCTTCAGAC
GAPDH	TTCAGCTCTGGGATGACCTT	CTCATGACCACAGTCCATGC

## Data Availability

Not applicable.

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
