# Peer review of "Effect of Novel AKT Inhibitor Vevorisertib as Single Agent and in Combination with Sorafenib on Hepatocellular Carcinoma in a Cirrhotic Rat Model"

_ijms, 2022, doi:10.3390/ijms232416206_

Round 1

Reviewer 1 Report

In this study, the authors examined the effect of novel AKT inhibitor Vevorisertib as single agent and in combination with Sorafenib on hepatocellular carcinoma in both cultured cells and a cirrhotic rat model. This study was performed accurately, and the results were reliable. However, there were several points to be addressed.

Major points:

1.       As the authors mentioned, AKT pathway is often activated in HCC cases and longer exposure to tyrosine kinase inhibitors such as sorafenib may lead to over-activation of the AKT pathway, leading to HCC resistance. In this study, AKT phosphorylation was evaluated only after single stimulation with Vevorisertib or Sorafenib in cultured cells (supplementary figure 2). To know the mechanisms for the combination therapy of these agents, AKT phosphorylation should be compared between single and combination treatments both in cultured cells and in HCC in rat liver samples.

2.      What is the mechanism for the normalization of vasculature and the reduction of fibrosis in veroisertib+sorafenib group? Is it due to or related to the AKT inhibition by veroisertib?

3.      Table 1 shows that body weight and liver weight were decreased and liver function was improved in veroisertib+sorafenib group. How the authors think the relationship between these changes and the changes in vasculature, fibrosis and tumor progression observed in veroisertib+sorafenib group? Did the authors quantify the food intake in veroisertib+sorafenib group?

Minor points:

1.       In the last paragraph of 2.2.1. of result section, the authors mentioned that vevorisertib and vevorisertib + sorafenib treatment do improve liver function without affecting glucose or triglyceride levels. However, the reviewer could not find the improvement of liver function in verorisertib group.

2.       In the figure legend for Fig. 2, A)~D) should be a)~d) as shown in the figure.

3.       Is combination therapy of vevorisertib + sorafenib already used in patients in clinical practice? Please mention it in the text, otherwise, readers of IJMS, who are not necessarily specialists of oncology, may not understand the importance of this study.

Author Response

Reviewer 1

In this study, the authors examined the effect of novel AKT inhibitor Vevorisertib as single agent and in combination with Sorafenib on hepatocellular carcinoma in both cultured cells and a cirrhotic rat model. This study was performed accurately, and the results were reliable. However, there were several points to be addressed.

Response: We thank Reviewer #1 for the positive feedback, her/his insights and very constructive comments.

 Major points:

  1. As the authors mentioned, AKT pathway is often activated in HCC cases and longer exposure to tyrosine kinase inhibitors such as sorafenib may lead to over-activation of the AKT pathway, leading to HCC resistance. In this study, AKT phosphorylation was evaluated only after single stimulation with Vevorisertib or Sorafenib in cultured cells (supplementary figure 2). To know the mechanisms for the combination therapy of these agents, AKT phosphorylation should be compared between single and combination treatments both in cultured cells and in HCC in rat liver samples.

Response: We thank Referee #1 for this criticism. We performed WB analysis on cell lines as suggested and included new Figure S3 “Supplementary Figure 3: Effect of Vevorisertib and Vevorisertib + Sorafenib treatment on phosphorylation of AKT. Representative western blot analysis of pAKT and AKT using HuH7 and HepG2 cell lines.”, showing that vevorisertib + Sorafenib treatment completely blocked AKT (Ser473) phosphorylation. Concerning the effect of combination treatment in vivo, we did not observe a significant difference in the pAKT/AKT levels in liver tissue between the treated groups. This was expected as the AKT inhibitor was administered in the schedule “5 days on and 9 days off” and animals were sacrificed 9 days after the last dose of Vevorisertib. New animal experiment would be necessary to reveal precisely the effect of AKT inhibitor (for instance withing 4, 24, 48h after the last dose of this AKT inhibitor).

  1. What is the mechanism for the normalization of vasculature and the reduction of fibrosis in veroisertib+sorafenib group? Is it due to or related to the AKT inhibition by veroisertib?

Response: We thank Referee #1 for this comment. Indeed, several markers seem to be additively improved by combination treatment while single treatment has lower efficacy. Sorafenib has been the backbone treatment of advanced-stage HCC for many years and we do observe a tendency for the normalization of vasculature and the reduction of fibrosis in Sorafenib single treatment in our rat model as well. Therefore we assume that the mechanism of veroisertib+sorafenib treatment efficacy is complex and can’t be fully attributed to AKT inhibitor.

One of the key players in the normalization of vasculature and the reduction of fibrosis is TGFβ. We observed that TGFβ expression is decreased in additive manner in the combination treatment. The discussion has been enriched by additional information addressing this point. “Moreover, it has been previously demonstrated that TGFβ activates the process of fibrogenesis in AKT dependent manner and that the fibrosis in a mouse model can be reduced by PI3K/mTOR inhibitor {Hettiarachchi, 2020 #1671}. Thus, we assume that the additive effect of the sorafenib and vevorisertib on reduction of TGFβ expression contributes to the reduction of fibrosis and to the normalization of vasculature in veroisertib+sorafenib treated group.”

  1. Table 1 shows that body weight and liver weight were decreased and liver function was improved in veroisertib+sorafenib group. How the authors think the relationship between these changes and the changes in vasculature, fibrosis and tumor progression observed in veroisertib+sorafenib group? Did the authors quantify the food intake in veroisertib+sorafenib group?

Response: We thank Referee #1 for this comment. As mentioned previously, the AKT inhibitor was administered in schedule “5 days on and 9 days off”. We observed the trend of body weight decrease at the end of the “5 days on” period, but the body weight loss was compensated during “9 days off” period. Overall, we observed no differences in food consumption. The absence of fibrosis and very small tumor size are indeed contributing to the decreased liver weight compared to vehicle-treated DEN-induced rats. The information about food consumption is now included in the revised version of manuscript.

 Minor points:

  1. In the last paragraph of 2.2.1. of result section, the authors mentioned that vevorisertib and vevorisertib + sorafenib treatment do improve liver function without affecting glucose or triglyceride levels. However, the reviewer could not find the improvement of liver function in verorisertib group.

Response: We apologize. The information is now corrected.

  1. In the figure legend for Fig. 2, A)~D) should be a)~d) as shown in the figure.

Response: The description has been corrected.

  1. Is combination therapy of vevorisertib + sorafenib already used in patients in clinical practice? Please mention it in the text, otherwise, readers of IJMS, who are not necessarily specialists of oncology, may not understand the importance of this study.

Response: We thank for this suggestion. The combination of vevorisertib + sorafenib was never used in clinical practice. We included this information in the discussion “It is important to note that the combination of velovorisertib + sorafenib has never been used in clinical practice and has even never been tested in pre-clinical models of CHC.”

Reviewer 2 Report

The authors demonstrate the antitumor effects of a novel Akt inhibitor in combination with sorafenib in a DEN-induced rat cirrhotic liver cancer model. The findings are necessary for the future development of new treatment approaches for hepatocellular carcinoma, and the paper could be considered acceptable with minor revisions. The following is a list of points requiring correction.

>For readers, supplemental table 1 and supplemental figure 2 should be cited in the text blow (p3, line 101-102 ): The western blot analyses indicated that vevorisertib treatment completely blocked AKT 101 (Ser473) phosphorylation in all tested cell lines, both at IC20 and IC50 concentrations.

>Table 1 contains the same data twice. And, there are two Table 1:one shows blood biochemistry tests in the result section and the other shows PCR primer sequences in the M&M section.

> Did you see any diarrhea or skin symptoms, which are often seen with Akt inhibitors? (Front Pharmacol. 2021, 12, 662232. doi: 10.3389/fphar.2021.662232.)

>If possible, it should be better to add the reference data of healthy control rat in Table 1.

> p4 line 136-137: The authors cite a decrease in aminotransferases as an improvement in liver function, but I disagree. Since elevated aminotransferase are often not seen in patients with cirrhosis, unlike hepatitis, a decrease in aminotransferase in a patient with cirrhosis does not mean that liver function has improved. Rather, elevated platelet counts, shortened PT, and increased serum albumin concentration are thought to be associated with improved liver function. I would agree that a decrease in bilirubin is indicative of improved liver function.

> Figure 2A-B uses uppercase letters (p4, line 144), but figure 2d-e uses lowercase letters. It is better to unify them. (p5, line 155 and 161). The legend of figure 3 and figure 4 has the same problems.

>The part in the discussion section (p8 line 222-243) should be described in the introduction section.

>In reference section, I found some error as below (some descriptions are incorrect or missing):

5. Rascio, F.; Spadaccino, F.; Rocchetti, M.T.; Castellano, G.; Stallone, G.; Netti, G.S.; Ranieri, E. The Pathogenic Role of PI3K/AKT Pathway in Cancer Onset and Drug Resistance: An Updated Review. Cancers 2021, 13, doi:10.3390/cancers13163949.

5. Rascio, F.; Spadaccino, F.; Rocchetti, M.T.; Castellano, G.; Stallone, G.; Netti, G.S.; Ranieri, E. The Pathogenic Role of PI3K/AKT Pathway in Cancer Onset and Drug Resistance: An Updated Review. Cancers 2021, 13, 3949; doi: 10.3390/cancers13163949. PMID: 34439105; PMCID: PMC8394096.

9. Marin, J.J.G.; Macias, R.I.R.; Monte, M.J.; Romero, M.R.; Asensio, M.; Sanchez-Martin, A.; Cives-Losada, C.; Temprano, A.G.; Espinosa-Escudero, R.; Reviejo, M., et al. Molecular Bases of Drug Resistance in Hepatocellular Carcinoma. Cancers 2020, 12, doi:10.3390/cancers12061663

9. Marin, J.J.G.; Macias, R.I.R.; Monte, M.J.; Romero, M.R.; Asensio, M.; Sanchez-Martin, A.; Cives-Losada, C.; Temprano, A.G.; Espinosa-Escudero, R.; Reviejo, M., et al. Molecular Bases of Drug Resistance in Hepatocellular Carcinoma. Cancers 2020, 12, 1663; doi: 10.3390/cancers12061663.

10. Shi, C.; Kwong, D.L.-W.; Li, X.; Wang, X.; Fang, X.; Sun, L.; Tang, Y.; Guan, X.-Y.; Li, S.-S. MAEL Augments Cancer Stemness Properties and Resistance to Sorafenib in Hepatocellular Carcinoma through the PTGS2/AKT/STAT3 Axis. Cancers 2022, 14, 2880

10. Shi C, Kwong DL, Li X, Wang X, Fang X, Sun L, Tang Y, Guan XY, Li SS. MAEL Augments Cancer Stemness Properties and Resistance to Sorafenib in Hepatocellular Carcinoma through the PTGS2/AKT/STAT3 Axis. Cancers 2022, 14, 2880; doi: 10.3390/cancers14122880.

20. Mroweh, M.; Roth, G.; Decaens, T.; Marche, P.N.; Lerat, H.; Macek Jílková, Z. Targeting Akt in Hepatocellular Carcinoma and Its Tumor Microenvironment. International journal of molecular sciences 2021, 22, 1794.

20. Mroweh M, Roth G, Decaens T, Marche PN, Lerat H, Macek Jílková Z. Targeting Akt in Hepatocellular Carcinoma and Its Tumor Microenvironment. Int J Mol Sci. 2021, 22, 1794. doi: 10.3390/ijms22041794.

23. Gungor, M.Z.; Uysal, M.; Senturk, S. The Bright and the Dark Side of TGF-β Signaling in Hepatocellular Carcinoma: Mechanisms, Dysregulation, and Therapeutic Implications. Cancers 2022, 14, 940.

23. Gungor, M.Z.; Uysal, M.; Senturk, S. The Bright and the Dark Side of TGF-β Signaling in Hepatocellular Carcinoma: Mechanisms, Dysregulation, and Therapeutic Implications. Cancers 2022, 14, 940; doi: 10.3390/cancers14040940.

25. Carreres, L.; Mercey-Ressejac, M.; Kurma, K.; Ghelfi, J.; Fournier, C.; Manches, O.; Chuffart, F.; Rousseaux, S.; Minoves, M.; Decaens, T., et al. Chronic Intermittent Hypoxia Increases Cell Proliferation in Hepatocellular Carcinoma. Cells 2022, 11, 2051.

25. Carreres, L.; Mercey-Ressejac, M.; Kurma, K.; Ghelfi, J.; Fournier, C.; Manches, O.; Chuffart, F.; Rousseaux, S.; Minoves, M.; Decaens, T., et al. Chronic Intermittent Hypoxia Increases Cell Proliferation in Hepatocellular Carcinoma. Cells. 2022, 11, 2051; doi: 10.3390/cells11132051.

Author Response

Reviewer 2

The authors demonstrate the antitumor effects of a novel Akt inhibitor in combination with sorafenib in a DEN-induced rat cirrhotic liver cancer model. The findings are necessary for the future development of new treatment approaches for hepatocellular carcinoma, and the paper could be considered acceptable with minor revisions. The following is a list of points requiring correction.

Response: We thank Reviewer #2 for appreciating our study and for her/his insights and constructive suggestions.

>For readers, supplemental table 1 and supplemental figure 2 should be cited in the text blow (p3, line 101-102 ): The western blot analyses indicated that vevorisertib treatment completely blocked AKT 101 (Ser473) phosphorylation in all tested cell lines, both at IC20 and IC50 concentrations.

Response: We thank for this suggestion. The information is now included for “Table S1, Figure S2”.

>Table 1 contains the same data twice. And, there are two Table 1:one shows blood biochemistry tests in the result section and the other shows PCR primer sequences in the M&M section.

Response: We apologize for these mistakes. The new version of manuscript is corrected.

> Did you see any diarrhea or skin symptoms, which are often seen with Akt inhibitors? (Front Pharmacol. 2021, 12, 662232. doi: 10.3389/fphar.2021.662232.)

Response: As we used the recommended schedule “5 days on and 9 days off”, we did not observe any diarrhea or skin symptoms in Akt inhibitor treated groups.

>If possible, it should be better to add the reference data of healthy control rat in Table 1.

Response: We thank for this excellent suggestion. We have information about a comparable biological analysis from healthy rats before the DEN injections (published previously in {Kurma, 2021 #1270} , Supplementary Table 2 ). The reference comparable data are now included in legend of Table 1.

> p4 line 136-137: The authors cite a decrease in aminotransferases as an improvement in liver function, but I disagree. Since elevated aminotransferase are often not seen in patients with cirrhosis, unlike hepatitis, a decrease in aminotransferase in a patient with cirrhosis does not mean that liver function has improved. Rather, elevated platelet counts, shortened PT, and increased serum albumin concentration are thought to be associated w ith improved liver function. I would agree that a decrease in bilirubin is indicative of improved liver function.

 Response: We thank for this comment. As recommended also by Reviewer 1, the sentence (previously p4 line 136-137) has been modified. Moreover, as Reviewer 2 recommended adding the reference data from healthy control rat, readers will have now much better possibility to evaluate the effect of treatment.

> Figure 2A-B uses uppercase letters (p4, line 144), but figure 2d-e uses lowercase letters. It is better to unify them. (p5, line 155 and 161). The legend of figure 3 and figure 4 has the same problems.

Response: We apologize. The new version of manuscript is corrected.

>The part in the discussion section (p8 line 222-243) should be described in the introduction section.

Response: We thank for this recommendation. We moved part of the discussion to the introduction section.

>In reference section, I found some error as below (some descriptions are incorrect or missing):

  1. Rascio, F.; Spadaccino, F.; Rocchetti, M.T.; Castellano, G.; Stallone, G.; Netti, G.S.; Ranieri, E. The Pathogenic Role of PI3K/AKT Pathway in Cancer Onset and Drug Resistance: An Updated Review. Cancers2021, 13, doi:10.3390/cancers13163949.

  1. Rascio, F.; Spadaccino, F.; Rocchetti, M.T.; Castellano, G.; Stallone, G.; Netti, G.S.; Ranieri, E. The Pathogenic Role of PI3K/AKT Pathway in Cancer Onset and Drug Resistance: An Updated Review. Cancers2021,13, 3949; doi: 10.3390/cancers13163949. PMID: 34439105; PMCID: PMC8394096.
  2. Marin, J.J.G.; Macias, R.I.R.; Monte, M.J.; Romero, M.R.; Asensio, M.; Sanchez-Martin, A.; Cives-Losada, C.; Temprano, A.G.; Espinosa-Escudero, R.; Reviejo, M., et al. Molecular Bases of Drug Resistance in Hepatocellular Carcinoma. Cancers2020, 12, doi:10.3390/cancers12061663

  1. Marin, J.J.G.; Macias, R.I.R.; Monte, M.J.; Romero, M.R.; Asensio, M.; Sanchez-Martin, A.; Cives-Losada, C.; Temprano, A.G.; Espinosa-Escudero, R.; Reviejo, M., et al. Molecular Bases of Drug Resistance in Hepatocellular Carcinoma. Cancers2020,12, 1663; doi: 10.3390/cancers12061663.
  2. Shi, C.; Kwong, D.L.-W.; Li, X.; Wang, X.; Fang, X.; Sun, L.; Tang, Y.; Guan, X.-Y.; Li, S.-S. MAEL Augments Cancer Stemness Properties and Resistance to Sorafenib in Hepatocellular Carcinoma through the PTGS2/AKT/STAT3 Axis. Cancers2022, 14, 2880

  1. Shi C, Kwong DL, Li X, Wang X, Fang X, Sun L, Tang Y, Guan XY, Li SS. MAEL Augments Cancer Stemness Properties and Resistance to Sorafenib in Hepatocellular Carcinoma through the PTGS2/AKT/STAT3 Axis. Cancers2022, 14, 2880; doi: 10.3390/cancers14122880.
  2. Mroweh, M.; Roth, G.; Decaens, T.; Marche, P.N.; Lerat, H.; Macek Jílková, Z. Targeting Akt in Hepatocellular Carcinoma and Its Tumor Microenvironment. International journal of molecular sciences2021, 22, 1794.

  1. Mroweh M, Roth G, Decaens T, Marche PN, Lerat H, Macek Jílková Z. Targeting Akt in Hepatocellular Carcinoma and Its Tumor Microenvironment. Int J Mol Sci.2021, 22, 1794. doi: 10.3390/ijms22041794.
  2. Gungor, M.Z.; Uysal, M.; Senturk, S. The Bright and the Dark Side of TGF-β Signaling in Hepatocellular Carcinoma: Mechanisms, Dysregulation, and Therapeutic Implications. Cancers2022, 14, 940.

  1. Gungor, M.Z.; Uysal, M.; Senturk, S. The Bright and the Dark Side of TGF-β Signaling in Hepatocellular Carcinoma: Mechanisms, Dysregulation, and Therapeutic Implications. Cancers2022, 14, 940; doi: 10.3390/cancers14040940.
  2. Carreres, L.; Mercey-Ressejac, M.; Kurma, K.; Ghelfi, J.; Fournier, C.; Manches, O.; Chuffart, F.; Rousseaux, S.; Minoves, M.; Decaens, T., et al. Chronic Intermittent Hypoxia Increases Cell Proliferation in Hepatocellular Carcinoma. Cells2022, 11, 2051.

  1. Carreres, L.; Mercey-Ressejac, M.; Kurma, K.; Ghelfi, J.; Fournier, C.; Manches, O.; Chuffart, F.; Rousseaux, S.; Minoves, M.; Decaens, T., et al. Chronic Intermittent Hypoxia Increases Cell Proliferation in Hepatocellular Carcinoma. Cells2022, 11, 2051; doi: 10.3390/cells11132051.

Response: We thank Referee #2 for this comment. The references have been updated.

Round 2

Reviewer 1 Report

In this revised version, the authors have performed several experiments and revised the manuscript sufficiently and conscientiously according to the reviewers’ suggestion.